# Progesterone Receptor Membrane Component 1 Regulates Cellular Stress Responses and Inflammatory Pathways in Chronic Neuroinflammatory Conditions

**DOI:** 10.3390/antiox13020230

**Published:** 2024-02-13

**Authors:** Seong-Lae Jo, Eui-Ju Hong

**Affiliations:** College of Veterinary Medicine, Chungnam National University, Daejeon 34134, Republic of Korea; jsr7093@o.cnu.ac.kr

**Keywords:** Alzheimer’s disease, neuroinflammation, progesterone receptor membrane component 1

## Abstract

Alzheimer’s disease (AD) is the leading cause of dementia and is one of the neurodegenerative diseases that are caused by neuronal death due to various triggers. Neuroinflammation plays a critical role in the development of AD. The neuroinflammatory response is manifested by pro-inflammatory cytokines, such as interleukin (IL)-1β, IL-6, and tumor necrosis factor-α; various chemokines; nitrous oxide; and reactive oxygen species. In this study, we evaluated the relevance of progesterone receptor membrane component 1 (PGRMC1), which is expressed in the brain cells during the induction of neuroinflammation. A lipopolysaccharide (LPS)-induced chronic neuroinflammation model and *Pgrmc1* knockdown cells were used to assess the inflammatory cytokine levels, AD-related factors, inflammation-related signaling, and cell death. *Pgrmc1* knockout (KO) mice had higher IL-1β levels after treatment with LPS compared with those of wild-type (WT) mice. Furthermore, *Pgrmc1* KO mice had higher levels of inflammatory factors, endoplasmic reticulum stress indicators, and AD-associated markers compared with those of WT mice who underwent LPS treatment or not. Finally, these indicators were observed in vitro using U373-MG astrocytes. In conclusion, the loss of PGRMC1 may promote neuroinflammation and lead to AD.

## 1. Introduction

Alzheimer’s disease (AD) is a degenerative condition that is caused by the death of nerve cells and is considered the leading cause of dementia [1]. A recent report suggested that 6.7 million Americans aged ≥65 years have AD in 2023, and the number may rise to approximately 13.8 million by 2060 [2]. The pathological features of AD include the accumulation of beta-amyloid plaque proteins on the outside of nerve cells and the building up of Tau protein strands inside nerve cells in the brain [1,3]. While early symptoms of AD include difficulty in remembering recent conversations, names, or events, later symptoms encompass difficulty in communication, disorientation, confusion, and poor judgment. Over time, patients with AD may have difficulty in speaking, swallowing, and walking. Ultimately, this progression can hinder the ability to stay active, making an individual vulnerable to physical complications that could lead to systemic inflammation [1,2]. The most significant risk factors for AD are age and genetics [4]. Furthermore, it was reported that traumatic brain injury [5], air pollution [6], and sleep deprivation [7] are risk factors for developing dementia. Interestingly, dementia can also be influenced by physical activity, smoking, diet, and blood pressure [8]. Recent reports have suggested that these modifiable factors can potentially prevent or delay the onset of dementia by up to 40% [9,10]. For example, low cardiovascular risk factors [11,12], physical activity [13], as well as social contact and cognitive engagement [9] may also play a promising role in the prevention of dementia.

Chronic inflammation contributes to AD development [2]. Recent studies have shown significant advances in understanding the link between neuroinflammation and AD [14,15]. Neuroinflammation is an inflammatory response in the central nervous system in response to various causes, such as injury, infection, traumatic brain injury, toxic metabolites, or autoimmunity [16,17]. The neuroinflammatory response is mediated by pro-inflammatory cytokines, such as interleukin (IL)-1β, IL-6, and tumor necrosis factor (TNF)-α; various chemokines; nitrous oxide; and reactive oxygen species [18,19,20]. Studies have shown that it is associated with increased levels of inflammatory cytokines, such as IL-1β, IL-6, and TNF-α, in both mice and patients with AD [21,22].

Progesterone receptor membrane component 1 (PGRMC1) is expressed in various tissues, including the liver, uterus, ovary, heart, and mammary gland, and in breast cancer [23,24,25,26,27]. It is located in the endomembranes, including the endoplasmic reticulum (ER), plasma membrane, nucleus, endosomes, Golgi apparatus, and cytoplasm [28]. PGRMC1 shares structural motifs with cytochrome b5 and is involved in drug, hormone, and lipid metabolism. Moreover, it has been linked to several functions [29]. For example, it interacts with progesterone to promote cell signaling [30] and has an impact on female reproductive function [31] as well as tumor proliferation [32] and energy metabolism [33]. When neuroinflammation was induced by lipopolysaccharides (LPS) in both *Pgrmc1* knockout (KO) and wild-type (WT) mice, we observed that the inflammatory cytokine levels in the brain and serum were higher in *Pgrmc1* KO mice compared to those of WT mice. Based on this evidence, we hypothesized that reduced PGRMC1 levels would be vulnerable to neuroinflammation, because the *Pgrmc1* KO mice exhibited higher levels of inflammation-related factors, such as nuclear factor (NF)-κB, ER stress, and apoptosis, compared to those of the WT mice. These findings will be reminiscent of our earlier observations in our alcoholic liver disease study [23]. In an early report concerning neurological diseases, PGRMC1 induced proliferation of neural progenitor cells, regulated neurogenesis and synapse remodeling [34,35], inhibited TNF-α induction of gene expression in neural cells [36], and had neuroprotective effects in neonatal hypoxic–ischemic brain injury [37], related to brain-derived neurotrophic factor (BDNF) signaling [38,39]. It has also been reported that PGRMC1 promotes the survival of human brain microvascular endothelial cells in AD and that the Sigma-2/PGRMC1 receptor mediates Abeta 42 oligomer binding and synaptic toxicity [40,41]. Therefore, we aimed to examine whether PGRMC1 activation could be an option for providing an innovative dementia treatment and developing prevention strategies.

## 2. Materials and Methods

### 2.1. Animals

*Pgrmc1* knockout (KO) mice were generated using TALEN with specific plasmids from ToolGen, Inc (Seoul, Republic of Korea). General methods for TALEN-mediated KO production were followed, involving a single dose of *Pgrmc1* TALEN mRNA (50 ng/µL) injected into C57BL/6 N mouse eggs [24]. We used 10-month-old wild-type mice and mice with a whole-body *Pgrmc1* KO. They were given a standard diet and had access to water, while being maintained on a 12 h light–dark cycle. All mouse experiments were approved and conducted in accordance with the guidelines of the Chungnam Facility Animal Care Committee (202209A-CNU-191). The mice were divided into four groups (*n* = 6 control group, *n* = 6 *Pgrmc1* knockout group, *n* = 6, lipopolysaccharide (LPS) group, *n* = 6 LPS + *Pgrmc1* knockout group) The sacrifices were performed under inhalational anesthesia with isoflurane. Mouse brains were cut in half on ice and collected using dissection tools in the following order: cerebellum, midbrain, hippocampus, thalamus, and cerebrum. Serum was allowed to clot for 30 min at room temperature after blood collection and then centrifuged (5000 rpm × 10 min) to obtain the supernatant.

### 2.2. Chronic Neuroinflammatory Mouse Modeling

We induced chronic neuroinflammatory states by imitating a published study [42]. Lipopolysaccharides (*E. coli* O111:B4, Sigma-Aldrich, St. Louis, MO, USA) were administered via intraperitoneal injection at a dosage of 750 μg/kg/body weight once every three days for a duration of one month. Eight hours prior to sacrifice, the mice were subjected to LPS treatment at a dosage of 1.5 mg/kg/body weight, while the control group received an equivalent dosage of distilled water.

### 2.3. Western Blotting

Protein samples from both the mouse cerebrum, U373-MG, and SH-SY5Y cells were extracted using a protein lysis buffer (T-PER reagent, 78510, Thermo Fisher Scientific Inc., Waltham, MA, USA). The protein concentration was determined using the Bradford assay with PRO-Measure solution (#21011, iNtRON Biotechnology, Kirkland, WA, USA). For electrophoresis, SDS-PAGE was performed on 10% or 12% polyacrylamide gels, depending on the protein size. The separated proteins were then transferred to a polyvinylidene fluoride (PVDF) membrane. To prevent non-specific binding, the membranes were blocked for 1 h using 3% Bovine Serum Albumin (BSA100, 9048-46-8, LPS solution, Daejeon, Republic of Korea) in diluted TBS-T buffer (04870517TBST4021, LPS solution). Primary antibodies were applied overnight at 4 °C, followed by washing the membranes with TBS-T. Subsequently, secondary antibodies were applied using the same procedure. The results were visualized using ECL solution (XLS025-0000, Cyanagen, Bologna, Italy), and Chemi Doc (Fusion Solo, VilberLourmat, Collégien, France) was used for imaging. All primary antibodies were diluted at a ratio of 1:2500 in 5% *w*/*v* BSA, while the secondary antibodies were diluted at the same ratio in 5% *w*/*v* skim milk. Detailed information on the primary and secondary antibodies is provided in Table 1.

### 2.4. Total RNA Extraction and Real-Time Quantitative PCR

The cerebral tissue and TRIzol reagent (15596-026, Life Technologies, Carlsbad, CA, USA) were placed in a 1.5 mL EP tube, homogenized, and subjected to centrifugation (13000 rpm, 10 min, 4 °C) after adding chloroform for phase separation. The upper aqueous phase was sequentially extracted to isolate RNA. To generate complementary DNA (cDNA) for reverse transcription, 3 µg of total RNA was reverse-transcribed using the reverse transcriptase kit (SG-cDNAS100, Smartgene, Daejeon, Republic of Korea) following the manufacturer’s protocol. Real-time PCR was performed using Excel Taq Q-PCR Master Mix (SG-SYBR-500, Smartgene, Daejeon, Republic of Korea) and Stratagene Mx3000P (Agilent Technologies, Santa Clara, CA, USA). Primers for real-time PCR were purchased from Bionics Inc. (Seoul, Republic of Korea), and primer information is shown in Table 2. We conducted experiments using the following protocol: In the first step, the temperature was maintained at 95 °C for 3 min. The second step involved 40 cycles, consisting of a denaturation stage at 95 °C for 15 s, an annealing stage at 60 °C for 15 s, and an extension stage at 72 °C for 30 s. Finally, the third step comprised a melt curve analysis, repeating denaturation at 95 °C for 10 s, annealing at 65 °C for 5 s, and an extension stage at 95 °C for 50 s. For accurate normalization, *Rplp0* was used as an internal control for samples. All experiments were performed three or more times, and mRNA expression levels were calculated based on cycle thresholds and assessed using amplification curves.

### 2.5. Serum IL-1β Levels

The mouse serum samples were diluted at 1:5 with PBS. The serum levels of IL-1β were used with colorimetric kit (RK00006, ABclonal, Woburn, MA, USA) and measured according to the manufacturer’s protocol.

### 2.6. Cell Culture and Gene Knockdown

The U373 MG and SH-SY5Y cell lines were purchased from the Korean Cell Line Bank (Seoul, Republic of Korea). All cells were maintained in DMEM (Welgene, LM001-05) supplemented with 5% (vol/vol) fetal bovine serum and 1% (vol/vol) penicillin–streptomycin in cell culture plates and incubated at 37 °C in a 5% CO_2_ atmosphere. In the experiment, the LPS treatment group was supplemented with a final concentration of 1 μg/mL, while the control group received supplementation with distilled water. The SC 79, a known activator of Akt phosphorylation, was supplemented to the SC 79 treatment group at a final concentration of 10 μM, dissolved in dimethyl sulfoxide (DMSO), and the control group for these treatments was also supplemented with DMSO. To perform *Pgrmc1* knockdown, siRNA transfection was carried out on a 12-well cell culture plate using Opti-MEM (11058021, Thermo Fisher Scientific Inc, Waltham, MA, USA) and Lipofectamine 2000 reagents (11668-027, Thermo Fisher Scientific Inc, Waltham, MA, USA). After washing the plate with DPBS, 500 μL of Opti-MEM was added to each well and incubated at 37 °C for 10 min in a 5% CO_2_ incubator. Negative control siRNA, as well as PGRMC1 siRNA #1 and #2, obtained from Bionics Co., Ltd. (Seoul, Republic of Korea), were added to Opti-MEM along with Lipofectamine. Following a six-hour incubation period, DMEM supplemented with 5% FBS and 1% P/S was added. Samples were collected after 72 h. The sequences for PGRMC1 siRNA #1 and #2 were 5′-CAGUACACAGUCA-3′ and 5′-CAGUACUCAAGUCAU-3′, respectively. All cell experiments were repeated at least 3 times.

### 2.7. Immunohistochemistry

For immunohistochemistry staining, 4 μm sections of paraffin-embedded tissues were affixed to silane-coated slides. The slides were alternately immersed in xylene baths twice, each for 5 min. Subsequently, the slides were immersed in 100%, 90%, 80%, and 70% ethanol for 3 min each. Following sequential rinsing in distilled water for 10 min, an Antigen Retrieval reagent was applied using an autoclave at 95 °C for 1 h to remove cross-links and reveal the concealed antigens. After allowing for sufficient cooling, the slides were washed with TBS-T. Tissue slides were then blocked with 3% BSA (9048-46-8, LPS solution, Daejeon, Republic of Korea) for 1 h. Primary antibodies (A19058, Company ABclonal Inc., Woburn, MA, USA) were incubated overnight at 4 °C. Following three washes with TBS-T, Alexa-Fluor secondary antibodies (#A21207, Thermo Fisher Scientific Inc., Waltham, MA, USA) were incubated overnight at 4 °C. Following three washes with TBS-T, the slides were mounted in ProLong Gold antifade reagent (P36935, Thermo Fisher Scientific Inc., Waltham, MA, USA). The stained slides were examined using a DMi8 microscope (Leica Microsystems, Wetzlar, Germany).

### 2.8. Statistical Analysis

All data were presented as the mean ± standard deviation (S.D). To determine the differences between means, we conducted a one-way analysis of variance (ANOVA), followed by a Tukey’s post-analysis, which were performed using GraphPad Prism Version 8 Software (GraphPad Inc., San Diego, CA, USA).

## 3. Results

### 3.1. PGRMC1 Is Located in the Brain

To investigate the expression of PGRMC1 protein in the mouse brain, we conducted a study to determine its presence and subcellular localization using Western blot. We focused on specific parts (cerebrum, thalamus, hippocampus, midbrain, cerebellum) of the mouse brain (Appendix A). Our results showed the presence of PGRMC1 protein by Western blot in the mouse brain, providing evidence for its expression in this tissue (Appendix A). In particular, a high expression of PGRMC1 was observed specifically in the cerebrum among the sites examined. Therefore, our study was focused on the cerebrum.

### 3.2. Pgrmc1 KO Mice Are Vulnerable to AD and Neuroinflammation

We showed that PGRMC1 protein exists in the mouse brain. To determine whether AD-associated proteins change in the presence or absence of PGRMC1 protein and whether PGRMC1 protein is vulnerable to neuroinflammation, we performed experiments in a neuroinflammation model (Figure 1A). In the neuroinflammatory model, LPS was administered for 1 month to establish a chronic inflammatory state, and high concentrations of LPS were administered for 8 h before sacrifice to enhance the response of inflammatory factors. First, we measured neuroinflammation markers. Interestingly, the serum IL-1β levels were not different between the WT and *Pgrmc1* KO mouse groups. However, the serum IL-1β levels significantly increased (*p* < 0.01, 1.27-fold) in the LPS-treated *Pgrmc1* KO mouse group compared to those in the LPS-treated WT mouse group. Moreover, after LPS treatment, WT (*p* < 0.001, 6.15-fold) and *Pgrmc1* KO (*p* < 0.01, 1.27-fold) mice showed high serum IL-1β levels compared to the normal state (Figure 1B). The *Il-1b* and *Tnf-α* mRNA levels were not different between the WT and *Pgrmc1* KO mouse groups. However, the *Il-1b* (*p* < 0.001, 2.50-fold) and *Tnf-α* (*p* < 0.001, 1.82-fold) mRNA levels significantly increased in the LPS-treated *Pgrmc1* KO mouse group compared to those in the LPS-treated WT mouse group. After LPS treatment, the WT (*Il-1b*: *p* < 0.001, 5.57-fold, *Tnf-α*: 2.24-fold) and *Pgrmc1* KO (*Il-1b*: *p* < 0.001, 2.50-fold, *Tnf-α*: 1.82-fold) mice showed high mRNA levels compared to the normal state (Figure 1C). The *Il-6* mRNA levels significantly increased (*p* < 0.01, 1.55-fold) in *Pgrmc1* KO mice compared to those in WT mice (Figure 1C). Moreover, the *Il-6* mRNA levels significantly increased (*p* < 0.001, 3.10-fold) in the LPS-treated *Pgrmc1* KO mouse group compared to those in the LPS-treated WT mouse group. After LPS treatment, the WT (*p* < 0.05, 1.46-fold) and *Pgrmc1* KO (*p* < 0.001, 2.64-fold) mice showed high mRNA levels compared to the normal state (Figure 1C).

Next, we measured the AD-associated markers. The amyloid-beta precursor protein (APP) levels were not different between the WT and *Pgrmc1* KO mouse groups. However, the App levels significantly increased (*p* < 0.001, 2.46-fold) in the LPS-treated *Pgrmc1* KO mouse group compared to those in the LPS-treated WT mouse group. A group of LPS-treated *Pgrmc1* KO mice exhibited increased (*p* < 0.001, 2.62-fold) APP protein levels compared to the normal state (Figure 1D). The Amyloid-beta (β-Amyloid) protein levels significantly increased (*p* < 0.05, 1.94-fold) in the *Pgrmc1* KO mice compared to those in the WT mice (Figure 1D). Moreover, the β-Amyloid protein levels significantly increased (*p* < 0.001, 1.30-fold) in the LPS-treated *Pgrmc1* KO mice compared to those in the LPS-treated WT mice. After LPS treatment, the WT (*p* < 0.001, 5.99-fold) and *Pgrmc1* KO (*p* < 0.001, 4.02-fold) mice showed high β-Amyloid protein levels compared to the normal state (Figure 1D). The Tau protein levels significantly increased (*p* < 0.001, 1.98-fold) in the *Pgrmc1* KO mice compared to those in the WT mice (Figure 1D). Further, the Tau protein levels significantly increased (*p* < 0.001, 1.44-fold) in the LPS-treated *Pgrmc1* KO mice compared to those in the LPS-treated WT mice. After LPS treatment, the WT (*p* < 0.001, 2.21-fold) and *Pgrmc1* KO (*p* < 0.001, 1.61-fold) mice showed high Tau protein levels compared to the normal state (Figure 1D). The PGRMC1 protein levels were significantly higher (*p* < 0.001, 25.6-fold) in the WT mice compared to those in the *Pgrmc1* KO mice (Figure 1D). The PGRMC1 protein levels were significantly higher (*p* < 0.001, 11.2-fold) in the LPS-treated WT mice compared to those in the *Pgrmc1* KO mice. Moreover, a group of LPS-treated WT mice exhibited decreased (*p* < 0.001, 50%) PGRMC1 protein levels compared to the normal state (Figure 1D).

### 3.3. Pgrmc1 KO Mice Aggravate Cell Death via NF-κB Upregulation

We have shown that the *Pgrmc1* KO mice are susceptible to LPS-induced neuroinflammation. It has been reported that the activation of the NF-κB pathway is mediated by increases in the pro-inflammatory cytokines TNF-α, IL-1β, and IL-6 [43]. In fact, we measured NF-κB-related signaling to examine whether NF-κB was activated in the absence of *Pgrmc1*.

The phospho-IκBα (pIκBα) protein levels were not different between the WT and *Pgrmc1* KO mouse groups. However, the pIκBα protein levels significantly increased (*p* < 0.001, 1.25-fold) in the LPS-treated *Pgrmc1* KO mouse group compared to those in the LPS-treated WT mouse group. Moreover, after LPS treatment, the WT (*p* < 0.001, 1.35-fold) and *Pgrmc1* KO (*p* < 0.001, 1.25-fold) mice showed high pIκBα protein levels compared to the normal state (Figure 2A). There was no difference in the IκBα protein levels between the WT and *Pgrmc1* KO mouse groups. However, after LPS treatment, the WT (*p* < 0.001, 64%) and *Pgrmc1* KO (*p* < 0.001, 63%) mice exhibited reduced IκBα protein levels compared to the normal state (Figure 2A). The levels of the pIκBα/IκBα ratio were not significantly different between the WT and *Pgrmc1* KO mouse groups. However, the pIκBα/IκBα ratio levels significantly increased (*p* < 0.01, 1.27-fold) in the LPS-treated *Pgrmc1* KO mice compared to those in the LPS-treated WT mice (Figure 2A). Moreover, after LPS treatment, the WT (*p* < 0.001, 2.14-fold) and *Pgrmc1* KO (*p* < 0.001, 2.82-fold) mice showed high pIκBα/IκBα ratio levels compared to the normal state (Figure 2A).

Next, the phospho-NF-κB (pNF-κB) protein levels significantly increased (*p* < 0.001, 1.95-fold) in the *Pgrmc1* KO mice compared to those in the WT mice (Figure 2A). Additionally, the pNF-κB protein levels significantly increased (*p* < 0.01, 1.18-fold) in the LPS-treated *Pgrmc1* KO mice compared to those in the LPS-treated WT mice. After LPS treatment, the WT (*p* < 0.001, 2.44-fold) and *Pgrmc1* KO (*p* < 0.001, 1.48-fold) mice showed high pNF-κB protein levels compared to the normal state (Figure 2A). The NF-κB protein levels significantly decreased (*p* < 0.001, 70%) in the *Pgrmc1* KO mice compared to those in the WT mice (Figure 2A). Further, the NF-κB protein levels significantly decreased (*p* < 0.001, 53%) in the LPS-treated *Pgrmc1* KO mice compared to those in the LPS-treated WT mice. After LPS treatment, the WT (*p* < 0.001, 49%) and *Pgrmc1* KO (*p* < 0.001, 53%) mice exhibited reduced NF-κB protein levels compared to the normal state (Figure 2A). The pNF-κB/NF-κB ratio levels significantly increased (*p* < 0.01, 2.61-fold) in the *Pgrmc1* KO mice compared to those in the WT mice (Figure 2A). Moreover, the pNF-κB/NF-κB ratio levels significantly increased (*p* < 0.001, 2.05-fold) in the LPS-treated *Pgrmc1* KO mice compared to those in the LPS-treated WT mice. After LPS treatment, the WT (*p* < 0.001, 5.43-fold) and *Pgrmc1* KO (*p* < 0.001, 1.99-fold) mice showed high pNF-κB/NF-κB ratio levels compared to the normal state (Figure 2A).

Our results showed that LPS induces neuroinflammation. It is also known that cell death occurs in cases of chronic inflammatory conditions [44]. Therefore, we measured apoptosis-related markers. The cleaved PARP protein levels significantly increased (*p* < 0.001, 3.00-fold) in the *Pgrmc1* KO mice compared to those in the WT mice (Figure 2B). In addition, the cleaved PARP protein levels significantly increased (*p* < 0.01, 1.80-fold) in the LPS-treated *Pgrmc1* KO mice compared to those in the LPS-treated WT mouse groups. After LPS treatment, the WT (*p* < 0.001, 4.76-fold) and *Pgrmc1* KO (*p* < 0.001, 1.77-fold) mice showed high cleaved PARP protein levels compared to the normal state (Figure 2B). The cleaved caspase-3 protein levels were not different between the WT and *Pgrmc1* KO mouse groups. However, the cleaved caspase-3 protein levels significantly increased (*p* < 0.001, 3.91-fold) in the LPS-treated *Pgrmc1* KO mouse group compared to those in the LPS-treated WT mouse group. A group of LPS-treated *Pgrmc1* KO mice exhibited increased (*p* < 0.001, 3.92-fold) cleaved caspase-3 protein levels compared to the normal state (Figure 2B). The cleaved caspase-3/caspase-3 ratio was not different between the WT and *Pgrmc1* KO mouse groups. However, the cleaved caspase-3/caspase-3 ratio significantly increased (*p* < 0.001, 5.67-fold) in the LPS-treated *Pgrmc1* KO mouse group compared to that in the LPS-treated WT mouse group. A group of LPS-treated *Pgrmc1* KO mice exhibited increased (*p* < 0.001, 9.85-fold) cleaved caspase-3/caspase-3 ratio levels compared to the normal state (Figure 2B).

In addition, it is known that the expression of glial fibrillary acidic protein (GFAP) is increased in situations such as brain injury, inflammation, and disease [45]. Since our results showed that NF-κB and apoptosis-related proteins were increased in neuroinflammatory conditions in the *Pgrmc1* KO mice compared to the WT, we observed the GFAP expression in the cerebrum region using IHC staining to confirm this (Appendix A). When the cerebrum was stained with the GFAP antibody, the positive signals were significantly increased (*p* < 0.05, 1.27-fold) in the *Pgrmc1* KO mice compared to those in the WT mice. In addition, the positive signals were significantly increased (*p* < 0.01, 1.22-fold) in the LPS-treated Pgrmc1 KO mice compared to those in the LPS-treated WT mouse group. After LPS treatment, the WT (*p* < 0.001, 1.59-fold) and *Pgrmc1* KO (*p* < 0.001, 1.53-fold) mice showed high positive signals compared to the normal state (Appendix A).

### 3.4. The Loss of Pgrmc1 Activates ER Stress in the Mouse Brain

In a previous study, we showed that *Pgrmc1* KO mice had higher ER stress-related protein markers in the liver compared to those of WT mice [23]. Moreover, several reports have suggested that inflammatory cytokines can induce ER stress, which can activate the unfolded protein response (UPR) [46,47]. We investigated whether ER stress increased during neuroinflammation or whether the absence of *Pgrmc1* led to increased ER stress.

The GRP78 protein levels significantly increased (*p* < 0.001, 2.14-fold) in the *Pgrmc1* KO mouse group compared to those in the WT mouse group (Figure 3). However, they were not different between the LPS-treated WT and the LPS-treated *Pgrmc1* KO mouse groups. A group of LPS-treated WT mice exhibited increased (*p* < 0.001, 3.80-fold) GRP78 protein levels compared to the normal state (Figure 3). The phospho-inositol-requiring enzyme type 1 (IRE1) alpha (pIRE1α) protein levels significantly increased (*p* < 0.001, 1.54-fold) in the *Pgrmc1* KO mouse group compared to those in the WT mouse group (Figure 3). Further, the pIRE1α protein levels significantly increased (*p* < 0.001, 1.95-fold) in the LPS-treated *Pgrmc1* KO mouse group compared those in the LPS-treated WT mouse group. After LPS treatment, the WT (*p* < 0.001, 1.96-fold) and *Pgrmc1* KO (*p* < 0.001, 2.49-fold) mice showed high pIRE1α protein levels compared to the normal state (Figure 3). The IRE1α protein levels significantly decreased (*p* < 0.001, 76%) in the *Pgrmc1* KO mice compared to those in the WT mice (Figure 3). However, the IRE1α protein levels were not different between the LPS-treated WT and LPS-treated *Pgrmc1* KO mouse groups (Figure 3). A group of LPS-treated WT mice exhibited decreased (*p* < 0.001, 77%) IRE1α protein levels compared to the normal state. The phospho-eIF2 alpha (peIF2α) protein levels significantly increased (*p* < 0.001, 2.25-fold) in the *Pgrmc1* KO mouse groups compared to those in the WT mouse groups (Figure 3). Additionally, the peIF2α protein levels significantly increased (*p* < 0.05, 1.20-fold) in the LPS-treated *Pgrmc1* KO mice compared to those in the LPS-treated WT mice. After LPS treatment, the WT (*p* < 0.001, 2.30-fold) and *Pgrmc1* KO (*p* < 0.01, 1.23-fold) mice showed high peIF2α protein levels compared to the normal state (Figure 3). However, the eIF2α protein levels were not significantly different between the groups (Figure 3). The activating transcription factor 6 (ATF6) protein levels significantly increased (*p* < 0.001, 1.42-fold) in the *Pgrmc1* KO mice compared to those in the WT mice (Figure 3). Furthermore, the ATF6 protein levels significantly increased (*p* < 0.001, 1.35-fold) in the LPS-treated *Pgrmc1* KO mouse group compared to those in the LPS-treated WT mouse group. After LPS treatment, the WT (*p* < 0.001, 2.06-fold) and *Pgrmc1* KO (*p* < 0.001, 1.96-fold) mice showed high ATF6 protein levels compared to the normal state (Figure 3). The ATF4 protein levels were not different between the WT and *Pgrmc1* KO mouse groups. However, after LPS treatment, the WT (*p* < 0.01, 1.23-fold) and *Pgrmc1* KO (*p* < 0.001, 1.41-fold) mice showed high ATF4 protein levels compared to the normal state (Figure 3). The C/EBP homologous protein (CHOP) protein levels significantly increased (*p* < 0.001, 2.27-fold) in the *Pgrmc1* KO mouse group compared to those in the WT mouse group (Figure 3). The CHOP protein levels significantly increased (*p* < 0.001, 1.36-fold) in the LPS-treated *Pgrmc1* KO mouse group compared to those in the LPS-treated WT mouse group. Finally, after LPS treatment, the WT (*p* < 0.001, 5.94-fold) and *Pgrmc1* KO (*p* < 0.001, 3.55-fold) mice showed high CHOP protein levels compared to the normal state (Figure 3).

### 3.5. The Reduced Expression of Pgrmc1 Leads to Increased Levels of AD-Related Proteins in LPS-Treated U373-MG Cells

Previous in vivo studies have shown that *Pgrmc1* KO mice have higher levels of AD-related proteins compared to those of WT mice and that they are vulnerable to AD after LPS treatment. We also showed that *Pgrmc1* expression was reduced in mice who were treated with LPS. Therefore, we decreased the expression of *Pgrmc1* in both astrocytes (U373-MG) and neurons (SH-SY5Y) and observed the subsequent changes following treatment with LPS. First, we observed the protein expression of PGRMC1 levels in each cell line. The protein expression of PGRMC1 was higher in U373-MG cells than that in SH-SY5Y cells (Appendix A).

To examine the changes in AD-related proteins when we reduced the expression of Pgrmc1, we measured the expression of AD-related proteins in U373-MG cells. The PGRMC1 protein levels significantly decreased (*p* < 0.001, 44%) in the *Pgrmc1* knockdown (KD) groups compared to those in the control groups (Figure 4A). Moreover, the PGRMC1 protein levels significantly decreased (*p* < 0.001, 64%) in the LPS-treated control groups compared to those in the LPS-treated *Pgrmc1* KD groups. A group of LPS-treated controls exhibited decreased (*p* < 0.001, 67%) PGRMC1 protein levels compared to the normal state (Figure 4A). The App levels significantly increased (*p* < 0.001, 2.98-fold) in the *Pgrmc1* KD groups compared to those in the control groups (Figure 4A). Further, the APP protein levels significantly increased (*p* < 0.001, 2.18-fold) in the LPS-treated *Pgrmc1* KD groups compared to those in the LPS-treated control groups. After LPS treatment, the control (*p* < 0.001, 1.61-fold) and *Pgrmc1* KD (*p* < 0.001, 1.17-fold) groups showed high APP protein levels compared to the normal state (Figure 4A). The β-amyloid levels significantly increased (*p* < 0.001, 3.77-fold) in the *Pgrmc1* KD group compared to those in the control group (Figure 4A). In addition, the β-amyloid protein levels significantly increased (*p* < 0.001, 1.71-fold) in the LPS-treated *Pgrmc1* KD group compared to those in the LPS-treated control group. After LPS treatment, the control (*p* < 0.001, 8.69-fold) and *Pgrmc1* KD (*p* < 0.001, 3.95-fold) groups showed high β-amyloid protein levels compared to the normal state (Figure 4A). The Tau protein levels significantly increased (*p* < 0.001, 2.07-fold) in the *Pgrmc1* KD group compared to those in the control group (Figure 4A). Moreover, the Tau protein levels significantly increased (*p* < 0.001, 1.68-fold) in the LPS-treated *Pgrmc1* KD group compared to those in the LPS-treated control group. A group of LPS-treated controls exhibited increased (*p* < 0.001, 1.34-fold) Tau protein levels compared to the normal state (Figure 4A).

Next, we monitored AD-related proteins in SH-SY5Y cells under the condition of *Pgrmc1* KD. The PGRMC1 protein levels significantly decreased (*p* < 0.001, 66%) in the *Pgrmc1* KD group compared to those in the control group (Appendix A). Additionally, the PGRMC1 protein levels significantly decreased (*p* < 0.001, 80%) in the LPS-treated control group compared to those in the LPS-treated *Pgrmc1* KD group. A group of LPS-treated controls exhibited decreased (*p* < 0.001, 78%) PGRMC1 protein levels compared to the normal state (Appendix A). The APP protein did not differ between the control and *Pgrmc1* KD groups, but the APP protein levels increased in the LPS-treated control (*p* < 0.001, 2.23-fold) and *Pgrmc1* KD (*p* < 0.001, 1.99-fold) groups when compared to the normal state. The β-amyloid and Tau levels were not significantly different between the groups (Appendix A). The proteins that are involved in NF-κB signaling and apoptosis in SH-SY5Y cells are presented in Appendix A.

In our data, when observing the expression of PGRMC1 in neurons and astrocytes, we found high PGRMC1 expression in astrocytes. Astrocytes, which constitute approximately 25% of the cerebral volume and are the most abundant glial cells in the nervous system, play a crucial role in AD [48]. They have neuroprotective functions, slowing plaque accumulation through Aβ clearance [49,50], and can be stimulated by microglia that are activated by noxious stimuli or by Aβ to increase β-amyloid production [51]. In response to inflammation, astrocytes secrete pro-inflammatory cytokines, such as IL-1β and TNF-α, to regulate the response [52,53]. Moreover, NF-κB, which is involved in the regulation of the inflammatory response [54,55], is a key signaling pathway in astrocytes that are involved in inflammation in AD cases. Therefore, in this study, we focused on astrocytes.

We measured NF-κB signaling and apoptosis-related protein. The pIκBα/IκBα ratio levels were not significantly different between the groups (Figure 4B). The pNF-κB/NF-κB ratio levels significantly increased (*p* < 0.001, 1.31-fold) in the *Pgrmc1* KD group compared to those in the control group (Figure 4B). The pNF-κB/NF-κB ratio levels significantly increased (*p* < 0.05, 1.11-fold) in the LPS-treated *Pgrmc1* KD group compared to those in the LPS-treated control group. After LPS treatment, the control (*p* < 0.001, 1.66-fold) and *Pgrmc1* KD (*p* < 0.001, 1.43-fold) groups showed high pNF-κB/NF-κB ratio levels compared to the normal state (Figure 4B). The cleaved PARP protein levels significantly increased (*p* < 0.001, 1.50-fold) in the *Pgrmc1* KD group compared to those in the control group (Figure 4B). Further, the cleaved PARP protein levels significantly increased (*p* < 0.001, 1.26-fold) in the LPS-treated *Pgrmc1* KD group compared to those in the LPS-treated control group. After LPS treatment, the control (*p* < 0.001, 1.47-fold) and *Pgrmc1* KD (*p* < 0.001, 1.24-fold) groups showed high cleaved PARP protein levels compared to the normal state (Figure 4B). The cleaved caspase3/capase3 ratio levels significantly increased (*p* < 0.001, 1.44-fold) in the LPS-treated *Pgrmc1* KD group compared to those in the LPS-treated control group. After LPS treatment, the control (*p* < 0.001, 2.54-fold) and *Pgrmc1* KD (*p* < 0.001, 2.21-fold) groups showed high cleaved caspase3/capase3 ratio levels compared to the normal state (Figure 4B).

### 3.6. The Reduced Expression of Pgrmc1 Upregulates ER Stress in U373-MG Cells

We measured the expression of ER stress-related proteins. The GRP78 protein levels significantly increased (*p* < 0.001, 2.50-fold) in the *Pgrmc1* KD group compared to those in the control group (Figure 5). Moreover, the GRP78 protein levels significantly increased (*p* < 0.001, 1.21-fold) in the LPS-treated *Pgrmc1* KD group compared to those in the LPS-treated control group. After LPS treatment, the control (*p* < 0.001, 3.26-fold) and *Pgrmc1* KD (*p* < 0.001, 1.58-fold) groups showed high GRP78 protein levels compared to the normal state (Figure 5). The pIRE1α protein levels significantly increased (*p* < 0.001, 2.09-fold) in the *Pgrmc1* KD group compared to those in the control group (Figure 5). In addition, the pIRE1α protein levels significantly increased (*p* < 0.001, 1.41-fold) in the LPS-treated *Pgrmc1* KD group compared to those in the LPS-treated control group. After LPS treatment, the control (*p* < 0.001, 2.75-fold) and *Pgrmc1* KD (*p* < 0.001, 1.41-fold) groups showed high pIRE1α protein levels compared to the normal state (Figure 5). The IRE1α, peIF2α, and eIF2α protein levels were not significantly different between the groups (Figure 5). The ATF6 protein levels significantly increased (*p* < 0.001, 5.99-fold) in the *Pgrmc1* KD group compared to those in the control group (Figure 5). Additionally, the ATF6 protein levels significantly increased (*p* < 0.001, 5.90-fold) in the LPS-treated *Pgrmc1* KD group compared to those in the LPS-treated control group. A group of LPS-treated *Pgrmc1* KD exhibited increased (*p* < 0.01, 1.05-fold) ATF6 protein levels compared to the normal state (Figure 5). The ATF4 protein levels significantly increased (*p* < 0.001, 4.60-fold) in the *Pgrmc1* KD group compared to those in the control group (Figure 5). Furthermore, the ATF4 protein levels significantly increased (*p* < 0.001, 2.55-fold) in the LPS-treated *Pgrmc1* KD group compared to those in the LPS-treated control group. After LPS treatment, the control (*p* < 0.001, 3.42-fold) and *Pgrmc1* KD (*p* < 0.001, 1.67-fold) groups showed high ATF4 protein levels compared to the normal state (Figure 5). The CHOP protein levels significantly increased (*p* < 0.001, 2.75-fold) in the *Pgrmc1* KD group compared to those in the control group (Figure 5). Finally, the CHOP protein levels significantly increased (*p* < 0.001, 1.11-fold) in the LPS-treated *Pgrmc1* KD group compared to those in the LPS-treated control group. After LPS treatment, the control (*p* < 0.001, 2.85-fold) and *Pgrmc1* KD (*p* < 0.001, 1.15-fold) groups showed high CHOP protein levels compared to the normal state (Figure 5). The proteins that are involved in ER stress in SH-SY5Y cells are shown in Appendix A.

### 3.7. Pgrmc1 Is a Regulator of Akt Levels, but Akt Is Not a Regulator of Pgrmc1

Our results suggest that a reduction in PGRMC1 levels activates ER stress. Surprisingly, ER stress is known to interact with Akt, and PGRMC1 regulates Akt activation [56,57,58], while PGRMC1 determines the Akt levels [37,59,60]. This evidence suggests that the upregulation of ER stress, achieved through a reduction in PGRMC1 levels, may be influenced by Akt. Consequently, we analyzed the Akt levels in the *Pgrmc1* KO mice and *Pgrmc1* KD cells to investigate the potential relationship between PGRMC1 and Akt.

First, we measured the change in the Akt protein levels in vivo. The pAkt protein levels significantly decreased (*p* < 0.001, 69%) in the *Pgrmc1* KO mouse group compared to those in the WT mouse group (Figure 6A). Moreover, the pAkt protein levels significantly decreased (*p* < 0.001, 74%) in the LPS-treated *Pgrmc1* KO mouse group compared to those in the LPS-treated WT mouse group. After LPS treatment, the WT (*p* < 0.001, 67%) and *Pgrmc1* KO (*p* < 0.01, 71%) groups exhibited reduced pAkt protein levels compared to the normal state (Figure 6A). However, the Akt protein levels were not significantly different between the groups. The pAkt/Akt ratio levels significantly decreased (*p* < 0.05, 71%) in the *Pgrmc1* KO mouse group compared to those in the WT mouse group (Figure 6A). In addition, the pAkt/Akt ratio levels significantly decreased (*p* < 0.05, 75%) in the LPS-treated *Pgrmc1* KO mouse group compared to those in the LPS-treated WT mouse group. After LPS treatment, the WT (*p* < 0.001, 68%) and *Pgrmc1* KO (*p* < 0.01, 71%) groups exhibited reduced pAkt/Akt ratio levels compared to the normal state (Figure 6A).

Next, we measured the change in the Akt protein levels in U373-MG cells. The pAkt protein levels significantly decreased (*p* < 0.001, 60%) in the *Pgrmc1* KD group compared to those in the control group (Figure 6B). Further, the pAkt protein levels significantly decreased (*p* < 0.001, 57%) in the LPS-treated *Pgrmc1* KD group compared to those in the LPS-treated control group. After LPS treatment, the control (*p* < 0.001, 90%) and *Pgrmc1* KD (*p* < 0.01, 60%) groups exhibited reduced pAkt protein levels compared to the normal state (Figure 6B). The Akt protein levels were not significantly different between the groups. The Akt protein levels were not different between the control and *Pgrmc1* KD mouse groups. However, after LPS treatment, the WT (*p* < 0.01, 2.91-fold) and *Pgrmc1* KD (*p* < 0.001, 2.94-fold) groups showed high Akt protein levels compared to the normal state (Figure 6B). The pAkt/Akt ratio levels significantly decreased (*p* < 0.001, 60%) in the *Pgrmc1* KD group compared to those in the control group (Figure 6B). Additionally, the pAkt/Akt ratio levels significantly decreased (*p* < 0.001, 56%) in the LPS-treated *Pgrmc1* KD group compared to those in the LPS-treated control group. After LPS treatment, the control (*p* < 0.001, 31%) and *Pgrmc1* KD (*p* < 0.01, 28%) groups exhibited reduced pAkt/Akt ratio levels compared to the normal state (Figure 6B). These results suggested that PGRMC1 and LPS may regulate Akt activation. Furthermore, it has not been investigated whether Akt is a regulator of PGRMC1. Therefore, we utilized SC 79, an Akt activator, to observe potential changes in PGRMC1 levels.

We observed SC 79-induced changes in PGRMC1 and Akt in the *Pgrmc1* KD state. The pAkt protein levels significantly decreased (*p* < 0.001, 52%) in the *Pgrmc1* KD group compared to those in the control group (Figure 6C). However, the pAkt protein levels were not different between the SC 79-treated control group and the SC 79-treated *Pgrmc1* KD group. After SC 79 treatment, the control (*p* < 0.001, 1.70-fold) and *Pgrmc1* KD (*p* < 0.001, 3.36-fold) groups showed high pAkt protein levels compared to the normal state (Figure 6C). The Akt protein levels were not significantly different between the groups. The pAkt/Akt ratio levels significantly decreased (*p* < 0.001, 46%) in the *Pgrmc1* KD group compared to those in the control group (Figure 6C). The pAkt/Akt ratio levels were not different between the SC 79-treated control group and the SC 79-treated *Pgrmc1* KD group (Figure 6C). Also, after SC 79 treatment, the control (*p* < 0.001, 1.75-fold) and *Pgrmc1* KD (*p* < 0.001, 3.27-fold) groups showed high pAkt/Akt ratio levels compared to the normal state (Figure 6C). The PGRMC1 protein levels significantly decreased (*p* < 0.001, 60%) in the *Pgrmc1* KD group compared to those in the control group (Figure 6C). However, after SC 79 exposure, the PGRMC1 protein levels showed no significant difference, indicating that SC 79 does not regulate the PGRMC1 level. Our results showed that, although Akt was activated by SC 79, PGRMC1 was not influenced by SC 79, indicating that Akt activation does not increase PGRMC1. In conclusion, PGRMC1 can regulate Akt, but Akt activation does not appear to regulate PGRMC1.

## 4. Discussion

To the best of our knowledge, this is the first study to examine whether PGRMC1 activation could be an option for dementia treatment. PGRMC1 is a protein that has been discussed as a potential therapeutic target, and many studies have shown that it can affect various aspects of disease through its multiple roles in many tissues and cells [23,27,29]. In neurology, PGRMC1 regulates the proliferation of neural progenitor cells, modulates synaptic remodeling, protects against hypoxic–ischemic brain injury, and protects against spinal cord-associated diseases or traumatic brain injuries. These findings suggest that PGRMC1 performs a variety of physiological and protective functions in the brain. This evidence presents PGRMC1 as one of the genes that can potentially treat neurodegenerative diseases [34,35,36,37,38,61]. Neurodegenerative diseases develop from various causes, with neuroinflammation being a key feature of AD. Inflammatory cytokines, such as IL-1β, IL-6, TNF-α, and transforming growth factor-β (TGF-β) have been found to increase the expression of amyloid precursor protein (APP), highlighting the relationship between neuroinflammation and neurodegeneration [52,62]. Therefore, we assessed whether *Pgrmc1* KO could affect neurodegeneration by inducing neuroinflammation with LPS.

Our results showed that the serum IL-1β levels, as well as the cerebral *Il-1b*, *Il-6*, and *Tnf* mRNA levels, were higher in *Pgrmc1* KO mice than those in WT mice after LPS-mediated inflammation. In 10-month-old *Pgrmc1* KO mice, the levels of IL-6 mRNA were also higher than those in WT mice without LPS-mediated inflammation. This suggests that the loss of *Pgrmc1* predisposes to neuroinflammation. When we measured the levels of AD-related markers to determine whether neuroinflammation affects AD development, the LPS-treated *Pgrmc1* KO mice exhibited higher levels of AD-related proteins than those of the WT mice. Furthermore, the *Pgrmc1* KO mice showed elevated levels of beta-amyloid and Tau proteins even under normal conditions. A similar pattern was observed in cells with *Pgrmc1* KD. According to previous reports [52,62,63], the induction of inflammatory cytokines in the brain is associated with AD and affects brain pathology. It can potentially influence memory and learning negatively by regulating APP production and processing and inducing changes in Tau proteins [18,64]. Based on the previous and current data, we suggest that PGRMC1 is vulnerable to neuroinflammation and AD.

Without LPS-mediated inflammation, the *Pgrmc1* KO mice exhibited higher levels of AD-related markers and IL-6 mRNA than the WT mice did. We hypothesized that these results could be attributed to the loss of *Pgrmc1*, contributing to the activation of NF-κB. Many studies have shown the potential importance of NF-κB in regulating disease susceptibility to various neurodegenerative diseases [65]. *Pgrmc1* inhibition enhances NF-κB signaling and NF-κB-mediated cytokine activation in a mouse model of hypoxic–ischemic brain injury [37]. Furthermore, we reported that the activation of the NF-κB pathway is facilitated by an increase in pro-inflammatory cytokines, which in turn contributes to neuroinflammation, microglial activation, oxidative stress complications, and, ultimately, cell death [66,67]. As expected, we observed that the loss of *Pgrmc1* amplified the NF-κB signaling pathway and apoptosis in LPS-mediated inflammation. Additionally, the *Pgrmc1* KO group showed a higher ratio of pNF-κB to NF-κB and cleaved PARP compared to the normal WT group. Furthermore, the level of inflammatory response in the mouse brain was assessed by IHC for the presence of GFAP. Although microglia and astrocytes are known to be key regulators of the inflammatory response [20], PGRMC1 was not expressed in microglia in the mouse brain [37], so we observed GFAP, which reveals astrocytes and inflammation. The IHC analysis revealed increased activity of the biomarker GFAP, which is associated with inflammation, neuronal damage, and cell death [45], in the *Pgrmc1* KO group compared to the normal WT group. This suggests that the loss of Pgrmc1 may be responsible for increased NF-κB activation and apoptosis, which could contribute to neuronal damage, a significant factor in the pathogenesis of AD.

The *Pgrmc1* KO mouse group exhibited higher levels of pNF-κB and cleaved PARP in their general state compared to the WT mice. These phenotypes are attributed to *Pgrmc1* deficiency, leading to the hypothesis that the interaction between *Pgrmc1* and ER stress may be involved. When ER stress is activated, the UPR pathway induces cellular apoptotic processes, activating NF-κB and pro-apoptotic pathways [68,69,70]. The activation of the ER stress sensor signaling pathways, including IRE1, protein kinase R-like endoplasmic reticulum kinase, and ATF6, is associated with NF-κB activation, which can result in increased production of inflammatory cytokines and cell death [71,72,73]. These interactions have been implicated in neurodegenerative diseases [74]. Previous studies on *Pgrmc1* and alcoholic liver disease have suggested that *Pgrmc1* KO mice experienced heightened ER stress [23]. Therefore, we expected to observe increased ER stress with the loss of *Pgrmc1* in the brain. As expected, the expression of GRP78, a key regulator of ER homeostasis, was elevated in *Pgrmc1* KO mice compared to that in WT mice. Given that ER stress interacts with inflammation signaling, we suggest that the loss of *Pgrmc1* affects AD by regulating both pathways.

Following a report that the inhibition of *Pgrmc1* with AG205 decreased the expression of BDNF, phosphoinositide 3-kinase (PI3K), and phosphorylated Akt levels [37], we observed a reduction in phospho-Akt in the *Pgrmc1* KO mice. Surprisingly, the *Pgrmc1* KO mice exhibited decreased pAkt/Akt ratio levels compared to those of the WT mice, and the above was similar in the *Pgrmc1* KD cells. Even though the phosphorylated expression of Akt increased after treatment with an Akt activator (SC 79), the PGRMC1 levels remained unchanged in the SC 79 group. Therefore, this suggests that PGRMC1 regulates Akt phosphorylation, apart from SC 79 activation.

Surprisingly, the regulation of Akt activation contributes to ER stress and cell death [75]. In the present study, the reduction in PGRMC1 levels decreased the activation of Akt and increased the activity of ER stress and cell death. Based on our findings, the increased phenotype of ER stress in *Pgrmc1* KO mice may be attributed to the influence of *Pgrmc1* on Akt regulation. Furthermore, we believe that inflammation-related signaling may have also increased as a result of elevated ER stress. Therefore, as ER stress and Akt interact [76], the relationship between PGRMC1 and ER stress can be further investigated.

Moreover, *Pgrmc1* was known to play a significant role in signal transduction and BDNF release in the brain [38,39]. The results of the present study showed that *Pgrmc1* regulated Akt phosphorylation, suggesting that *Pgrmc1* may contribute to the treatment of neurodegeneration through the BDNF/PI3K/Akt pathway by regulating Akt. Furthermore, the observed decrease in AKT’s response to LPS could be attributed to the potential induction of insulin resistance by LPS. We anticipate that these effects could lead to a reduction in AKT activity. Notably, one study reported that ligands activate the PI3K/AKT signaling pathway under normal conditions. However, in insulin-resistant brains, ER stress occurs, leading to impairment of the PI3K/AKT pathway [77,78]. Putting the results into perspective, the influence of LPS could induce ER stress and consequently downregulate the AKT pathway. These findings suggest a possible link between LPS-induced ER stress and the observed decrease in AKT activity, supporting the notion that ER stress may contribute to the modulation of the PI3K/AKT pathway.

In this study, we evaluated the correlation between PGRMC1 and neuroinflammation using a *Pgrmc1* KO mouse model and *Pgrmc1* KD astrocyte cells. In the absence of PGRMC1, there was an increase in cytokine activity during inflammatory states, and markers related to Alzheimer’s disease also showed an elevation. Therefore, the decrease in PGRMC1 is revealed to facilitate neurodegenerative diseases, which was attributed to the increased inflammatory response. When the reduction in *Pgrmc1* exhibited a pattern of activating ER stress and reducing Akt activity. Therefore, we confirmed the regulatory role of PGRMC1 in Akt, suggesting that PGRMC1 does, indeed, regulate Akt. A PGRMC1 reduction activates the NF-κB pathway and ER stress to promote cell death due to cytokine hyperactivity via the inhibition of Akt activity. These findings suggest that modulation of PGRMC1 may be considered in treating neurodegenerative diseases.

However, our study provided limited insights into the responsiveness of PGRMC1 in neuron cells. This is because the expression of PGRMC1 varies depending on the differentiation status of neuronal cell lines [40]. Since our experiments utilized undifferentiated neuronal cell lines, we propose that future research should investigate the relationship between PGRMC1 and neuroinflammation using differentiated neuronal cell lines to gain a more comprehensive understanding.

## 5. Conclusions

We found that the reduction in PGRMC1 levels activated the NF-κB and ER stress pathways, leading to an increase in indicators that are associated with neurodegenerative diseases due to pro-inflammatory cytokine elevation. In addition, the reduction in Pgrmc1 increased factors related to ER stress, influencing the decrease in AKT activation. This can be considered to contribute to the downregulation of the BDNF/PI3K/Akt pathway, thereby impacting neurodegenerative diseases. These observations suggest that the modulation of PGRMC1 may be proposed as a treatment and prevention strategy for neurodegenerative diseases.

## Figures and Tables

**Figure 1 antioxidants-13-00230-f001:**
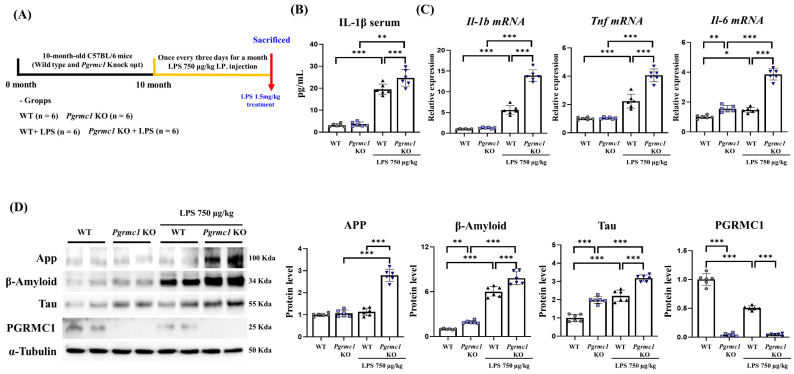
Vulnerability of the brains of progesterone receptor membrane component 1 (*Pgrmc1*) knockout (KO) mice to neuroinflammation and Alzheimer’s disease (AD). (**A**) Schematic representation of the experimental timeline. Twenty-four mice, aged 10 months, were divided into four groups. The lipopolysaccharide (LPS) treatment group received LPS at 750 μg/kg of body weight every 3 days via intraperitoneal injection for 1 month. Eight hours before sacrifice, mice were given LPS at 1.5 mg/kg of body weight, and control groups received the same volume of distilled water. The cerebrum was extracted from the brain. (**B**) Measurement of serum interleukin (IL)-1β levels after sacrifice. (**C**) Quantitative real-time polymerase chain reaction (qRT-PCR) analysis of *Il-1b*, *Tnf*, and *Il-6* gene mRNA levels in the cerebrum of each group of male mice. RPLP0 was used as an internal control. (**D**) Western blot analysis and quantification of neurodegenerative disease-related proteins and PGRMC1 protein levels in the cerebrum of mice. Alpha-Tubulin served as an internal control. Differences between means were assessed using one-way ANOVA, followed by Tukey’s post-analysis. Values represent means +/− standard deviation. * *p* < 0.05, ** *p* < 0.01, *** *p* < 0.001 (*n* = 6 in control WT and KO groups, *n* = 6 in LPS-treated WT and KO groups). All experiments were replicated at least 3 times.

**Figure 2 antioxidants-13-00230-f002:**
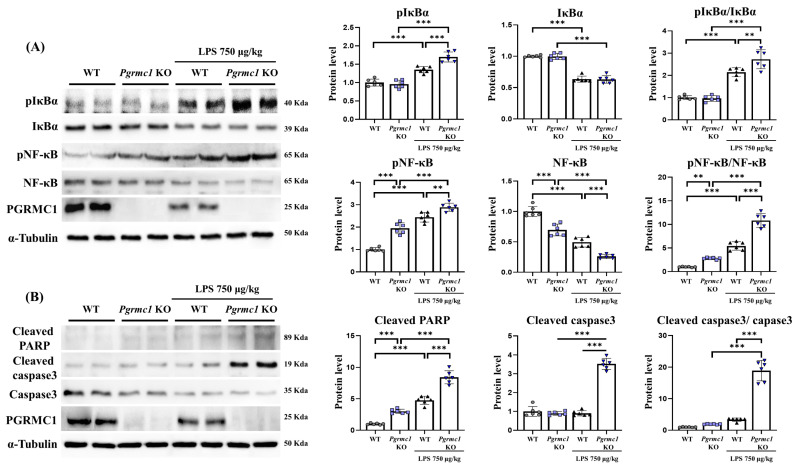
Loss of *Pgrmc1* exacerbates cell death via upregulating NF-κB. (**A**) Western blot analysis and quantification of inflammation-related proteins were evaluated in the cerebrum of each group of male mice. Alpha-Tubulin was used for internal control. (**B**) Western blot analysis and quantification of apoptosis-related proteins were evaluated in the cerebrum of each group of male mice. Alpha-Tubulin served as an internal control. Differences between means were assessed using one-way ANOVA, followed by Tukey’s post-analysis. Values represent means +/− standard deviation. ** *p* < 0.01, *** *p* < 0.001 (*n* = 6 in control WT and KO groups, *n* = 6 in LPS-treated WT and KO groups). All experiments were replicated at least 3 times.

**Figure 3 antioxidants-13-00230-f003:**
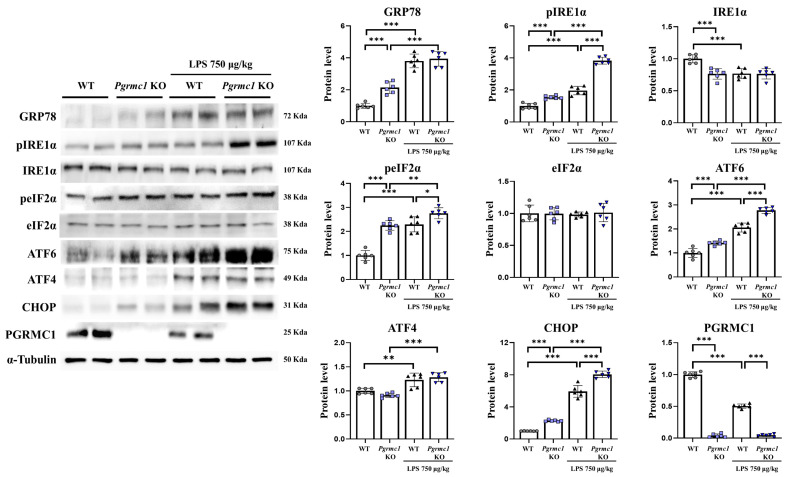
Loss of *Pgrmc1* increases endoplasmic reticulum (ER) stress-related markers in the mouse brain. Western blot analysis and quantification of ER stress-related proteins were evaluated in the cerebrum of each group of male mice. Alpha-Tubulin served as an internal control. Differences between means were assessed using one-way ANOVA, followed by Tukey’s post-analysis. Values represent means +/− standard deviation. * *p* < 0.05, ** *p* < 0.01, *** *p* < 0.001 (*n* = 6 in control WT and KO groups, *n* = 6 in LPS-treated WT and KO groups). All experiments were replicated at least 3 times.

**Figure 4 antioxidants-13-00230-f004:**
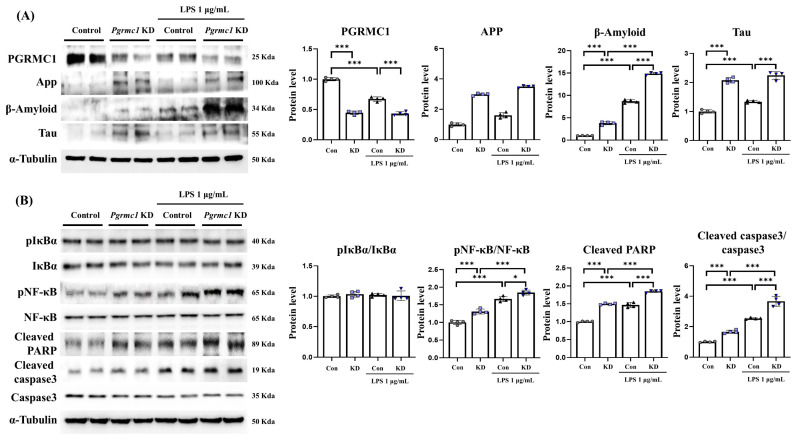
*Pgrmc1* downregulation increases levels of AD- and inflammation-related protein markers in LPS-treated U373-MG cells. (**A**) Western blot analysis and quantification of Alzheimer’s disease-related proteins were evaluated in the U373-MG cells. Alpha-Tubulin was used for internal control. (**B**) Western blot analysis and quantification of inflammation-related proteins were evaluated in the U373-MG cells. Alpha-Tubulin served as an internal control. Differences between means were assessed using one-way ANOVA, followed by Tukey’s post-analysis. Values represent means +/− standard deviation. * *p* < 0.05, *** *p* < 0.001 (*n* = 4 in control and KD groups, *n* = 4 in LPS-treated control and KD groups). All experiments were repeated at least 3 times.

**Figure 5 antioxidants-13-00230-f005:**
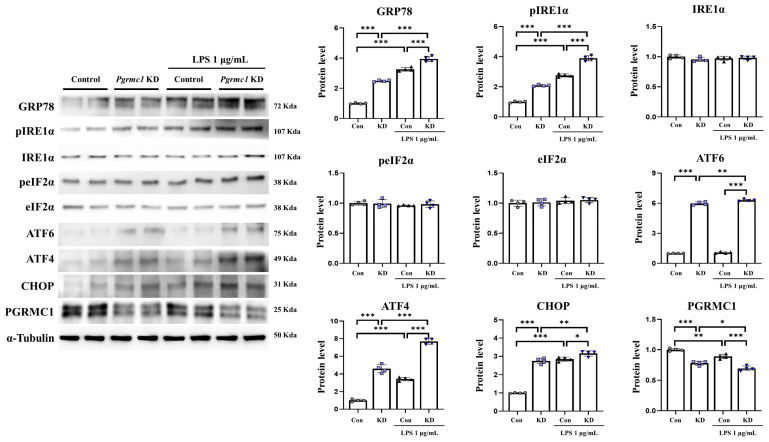
*Pgrmc1* downregulation increases levels of ER stress-related markers in U373 cells. Western blot analysis and quantification of ER stress-related proteins were evaluated in U373-MG cells. Alpha-Tubulin served as an internal control. Differences between means were assessed using one-way ANOVA, followed by Tukey’s post-analysis. Values represent means +/− standard deviation. * *p* < 0.05, ** *p* < 0.01, *** *p* < 0.001 (*n* = 4 in control and KD groups, *n* = 4 in LPS-treated control and KD groups). All experiments were repeated at least 3 times.

**Figure 6 antioxidants-13-00230-f006:**
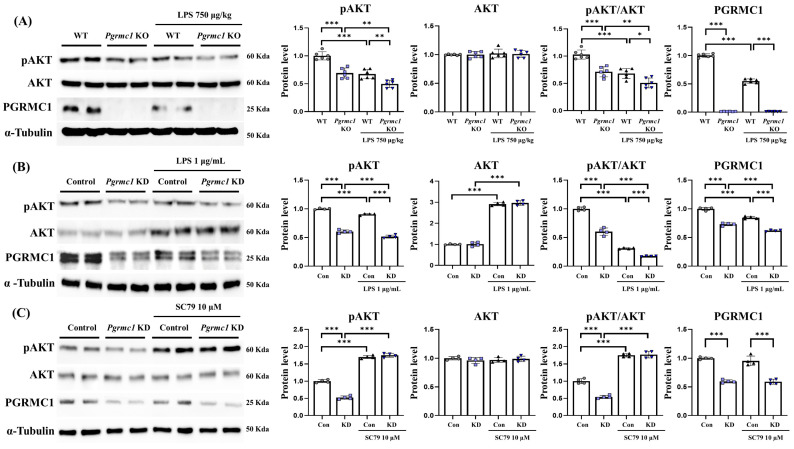
*Pgrmc1* regulates AKT activity. (**A**) In mouse brain, AKT and phospho-AKT protein levels in each group were evaluated by Western blot analysis and quantification. Alpha-tubulin was used as an internal control (*n* = 6 in control WT and KO groups, *n* = 6 in LPS-treated WT and KO groups). (**B**) In U373-MG cells, AKT and phospho-AKT protein levels in each group were evaluated by Western blot analysis and quantification. (*n* = 4 in control and KD groups, *n* = 4 in LPS-treated control and KD groups). (**C**) After treatment with AKT activator, Western blot analysis and quantification of AKT-related genes were evaluated upon SC 79 treatment in *Pgrmc1* KD U373-MG cells. Alpha-Tubulin was used for internal control (*n* = 4 in control and KD groups, *n* = 4 in SC 79-treated control and KD groups). Alpha-Tubulin served as an internal control. Differences between means were assessed using one-way ANOVA, followed by Tukey’s post-analysis. Values represent means +/− standard deviation. * *p* < 0.05, ** *p* < 0.01, *** *p* < 0.001.

**Table 1 antioxidants-13-00230-t001:** Primary and secondary antibodies used for Western blot.

Antibodies	Type	Lot.	Inc.
PGRMC1	Rabbit monoclonal	#13856	Cell signaling technology (Danvers, MA, USA)
Amyloid-beta	Mouse monoclonal	sc-28365	Santa Cruz biotechnology (Dallas, TX, USA)
Tau	Rabbit monoclonal	A1103	Company ABclonal, Inc. (Boston, MA, USA)
GRP78	Rabbit monoclonal	GTX113340	Genetex, Inc. (Irvine, CA, USA)
Phospho-IRE1α (Ser724)	Rabbit monoclonal	GTX63722	Genetex, Inc. (Irvine, CA, USA)
IRE1α	Rabbit monoclonal	ab37073	Abcam, Inc. (Boston, MA, USA)
Phospho-eIF2α (Ser52)	Rabbit monoclonal	#3597	Cell signaling technology (Danvers, MA, USA)
eIF2α	Rabbit monoclonal	#9722	Cell signaling technology (Danvers, MA, USA)
ATF4	Rabbit monoclonal	#11815	Cell signaling technology (Danvers, MA, USA)
ATF6	Rabbit monoclonal	ab65838	Abcam, Inc. (Boston, MA, USA)
CHOP	Mouse monoclonal	MA1-250	Invitrogen (Waltham, MA, USA)
Phospho-AKT (Ser473)	Rabbit monoclonal	#4058	Cell signaling technology (Danvers, MA, USA)
AKT	Rabbit monoclonal	#9272	Cell signaling technology (Danvers, MA, USA)
Phospho-IkBα (Ser32)	Rabbit monoclonal	#2859	Cell signaling technology (Danvers, MA, USA)
IkBα	Mouse monoclonal	#4814	Cell signaling technology (Danvers, MA, USA)
Phospho-NF-κB (Ser536)	Rabbit monoclonal	#3033	Cell signaling technology (Danvers, MA, USA)
NF-κB	Rabbit monoclonal	#8242	Cell signaling technology (Danvers, MA, USA)
Cleaved PARP	Rabbit monoclonal	#9544	Cell signaling technology (Danvers, MA, USA)
Caspase3	Rabbit monoclonal	#9665	Cell signaling technology (Danvers, MA, USA)
Cleaved caspase3	Rabbit monoclonal	#9664	Cell signaling technology (Danvers, MA, USA)
Alpha-Tubulin	Mouse monoclonal	66031-1-Ig	Proteintech Group Inc. (Rosemont, IL, USA)
Anti-Mouse IgG	Goat	121507	Jackson Immunoresearch (West Grove, PA, USA)
Anti-Rabbit IgG	Mouse	123213	Jackson Immunoresearch (West Grove, PA, USA)

**Table 2 antioxidants-13-00230-t002:** Primers used for real-time PCR.

Gene Name	Upper Primer (5′-3′)	Lower Primer (5′-3′)	Accession Number	Species
*Il-1b*	GAA ATG CCA CCT TTT GAC AGT G	CTG GAT GCT CTC ATC AGG ACA	NM_008361	Mouse
*Il-6*	AGT TGC CTT CTT GGG ACT GA	TCC ACG ATT TCC CAG AGA AC	NM_031168	Mouse
*Tnf*	CCT GTA GCC CAC GTC GTA G	GGG AGT AGA CAA GGT ACA ACC C	NM_013693	Mouse
*Rplp0*	GCA GCA GAT CCG CAT GTC GCT CCG	GAG CTG GCA CAG TGA CCT CAC ACG G	NM_007475	Mouse

## Data Availability

The data presented in this study are available in the article and Appendix A.

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
