# Peer review of "Progesterone Receptor Membrane Component 1 Regulates Cellular Stress Responses and Inflammatory Pathways in Chronic Neuroinflammatory Conditions"

_antioxidants, 2024, doi:10.3390/antiox13020230_

Round 1
Reviewer 1 Report
Comments and Suggestions for Authors
In this manuscript the authors describe the role of Progesterone receptor membrane component 1 (pgmrc1) in neuroinflammation, ER stress, apoptosis and marker proteins of Alzheimer's disease (AD).
The relationship between all these processes is well described in the literature. The novelty of the work seems to analyse the role of PGMRC1 in some of these processes in a mice model of neuroinflammation in vivo and in vitro. The in vivo experiments are well designed and performed and the analysis are very complete.
The manuscript needs an exhaustive review of the writing and presentation of results since it is not clear and contains various conceptual errors.
Some of the points to improve are:
-Rewrite the abstract. This sentence, for example, is not understood: "AD is closely associated with these triggers of neuroinflammation." Also, it says "brain cells and astrocytes", astrocytes are brain cells.
-The introduction describes many general data about Alzheimer's that are not necessary but does not describe the relationship of PGMRC1 with this disease (There are several publications on the matter, not cited by the authors), although it does describe its neuroprotective and anti-inflammatory role. In addition, some results are already included in the introduction that it would be better not to describe here, even if it is to justify the work.
-Methods:
The number of animals used in the study, six, is small.
In methods, it would be necessary to indicate in more detail the origin of the Knock down mouse.
Also define in the methods what SC79 is.
It would be necessary to better describe how the animals are euthanized and how the different areas are dissected, as well as the time at which the serum is extracted. Although this information is in other points of the manuscript.
The authors must justify the reason for designing the administration of LPS, during one month and a higher dose 8 hours before euthanizing the animals.
-The following paragraph "the subcellular localization of PGRMC1 was identified through immunohistochemical analysis, showing its distribution within specific cellular compartments in the brain (Supplemental Figure S1B)" is not true. It is a western blot and not immunohistochemistry and in addition the subcellular localization is not analyzed but rather the expression in different brain areas.
-In the text of the results it could be added where the different parameters of neuroinflammation and proteins related to AD, or ER stress, are being measured.
-This part is repeated but it also does not say the same thing both times: "the App levels significantly increased (p<0.05, 1.94-fold) in Pgrmc1 KO mice compared to those in WT mice (Figure1D). Furthermore, the App levels significantly increased (p<0.001, 1.30-fold) in LPS-treated Pgrmc1 KO mice compared to those in LPS-treated WT mice (Figure 1D)".
- This sentence is not correct either: "the PGMRC1 protein levels were significantly higher (p<0.001, 11.2-fold) in LPS-treated Pgrmc1 KO mice compared to those in LPS-treated WT mice (Figure 1D)."
-In Fig. 1 it indicates "protein level" on the y-axis although it shows mRNA levels.
-Why do NFkB levels decrease in the mutant and in WT mice treated with LPS? This must be commented.
- The expression: "AD or apoptosis-related genes" is repeated many times, but since the contents of the proteins are analyzed, it would be more accurate to say proteins related to...In addition, it would be better to specify in the legends of the figures which specific proteins are analyzed.
-Why is the expression of PGMRC1 showed in almost all the figures?
-Justify in vitro experiments with neurons and astrocytes.
-In general, the statistical differences between WT and WT treated with LPS in each protein analyzed should be added. For example, is the IRE or ATF4 protein altered by LPS administration?
-The results in neurons should be described correctly, even if they are in a supplementary figure. But furthermore, this decision is not clearly justified and is not discussed in the discussion. It would be necessary to comment on why it could be that the neurons were less affected or the influence of the fact that they are cell lines and not primary cells, which may be responsible for some results in the content of different proteins.
- -Why does the pNFkB content decrease in control neurons after LPS administration?
-In astrocytes the pIkB content does not change but in vivo it does. This indicates that the effect of PGMRC1 downregulation is different in vivo than in cultured astrocytes. Authors must discuss this.
-Why the levels of caspase-3 and NFkB (and IRE and eIF2) don't change in astrocytes as occurs in the cerebrum? It seems that the effects, and therefore the mechanisms of regulation of these processes by PGRMC1 are different. This should be discussed .
-This sentence is not correct: "the PGRMC1 protein levels were not different between the SC79-treated control group and the SC79-treated Pgrmc1 KD group"
-Does the AKT activator increase pAKT? Describe the statistics.
-Discuss the results, commenting on the effects of LPS on the WT, according to what is already known or not about the effects of LPS on these pathways.
Author Response
"Please see the attachment."

Reviewer 2 Report
Comments and Suggestions for Authors
In this study, the authors evaluated the relevance of progesterone receptor membrane component 1 (PGRMC1), which is expressed in the brain cells and astrocytes during the induction of neuroinflammation. A lipopolysaccharide (LPS)-induced chronic-neuroinflammation model and Pgrmc1 knock down cells were used to assess the inflammatory cytokine levels, Alzheimer's Disease (AD)-related factors, inflammation-related signaling, and cell death. Pgrmc1 knock out (KO) mice had higher IL-1β levels after treatment with LPS compared with those of wild-type (WT) mice. Furthermore, Pgrmc1 KO mice had higher levels of inflammatory factors, endoplasmic reticulum stress indicators, and AD-associated markers compared with those of WT mice who underwent LPS treatment or not. Finally, these indicators were observed in vitro using U373-MG astrocytes. The authors concluded that the loss of PGRMC1 may promote neuroinflammation and lead to AD. There are several comments on this manuscript.
1. The authors use LPS as a stimulator to simulate the inflammatory pathology of AD. It’s known that AD is a neurodegenerative disease characterized by extracellular deposition of Aβ plaques and intracellular accumulation of hyper-phosphorylated tau protein and NFTs (neurofibrillary tangles). Therefore, why not use Aβ40/Aβ42 as the inducible factor? Or else, the authors can only conclude that the loss of PGRMC1 may promote neuroinflammation and lead to neuroinflammation associated neuronal injury.
2. Given that microglia play crucial roles in the brain innate immune system, the increased inflammation in Pgrmc1 KO mouse brain may also be produced from activated microglia. Thus, immunofluorescent staining of microglia and/or astrocyte in Pgrmc1 KO mouse brain treated with or without LPS is suggested to be provided in the text. Additionally, how about the expression of PGRMC1 in microglia?
3. As the authors have found in the text, APP, β amyloid and Tau are increased in Pgrmc1 KO mouse brain in response to LPS treatment in figure 1. The authors did not provide which form of β amyloid or the phosphorylation of Tau.
4. Neural damage or cell death in Figure 2 can be further verified by assessing the expression of TUNEL staining. Furthermore, the authors’ conclusion that “loss of Pgrmc1 exacerbates cell death via upregulating NF-κB” is inadequate, as they did not use NF-κB inhibitor to demonstrate whether the NF-κB inhibitor can abolish cell death in Pgrmc1 KO mouse brain.
5. The authors try to correlate PGRMC1 with AD, they detect APP and Tau in U373-MG cells but not in neuron or SH-SY5Y cells. Actually, the pathological hallmarks of AD are in the extracellular (β amyloid) or intracellular (pTau) of neurons. Therefore, it’s important to detect those indexes in neurons or in APPswe695 neuronal cell lines. Additionally, how about the expression of PGRMC1 in AD brain? If the authors cannot detect it because of lacking AD samples, maybe they can find PGRMC1 expression in data base of AD.
Author Response
"Please see the attachment."

Round 2
Reviewer 2 Report
The authors have supplemented several experiments to answer the reviewer’s comments, which has improved the manuscript. However, there are still several comments on this manuscript.
1. Given the increased inflammation in PGRMC1 KO mouse in response to LPS, it’s easily to hypothesize that PGRMC1 may mainly from microglia or astrocyte in the brain. The authors provide the evidence that cellular expression of PGRMC1 is not expressed in microglia in response to reviewer. Please cite the reference to explain it in the discussion. Please put the figure of GFAP staining in response to review to Supplemental Figure. The immunofluorescent staining in Supplemental Figure S1A seems not specific.
2. It would be more readable and interesting if the authors detailed explain the possible mechanisms why the expressions of APP, Abeta, Tau are increased in PGRMC1 KO mouse brain in the discussion.
3. PI3K/Akt signaling pathway is also related to NF-kB activation. In this manuscript, the authors found that pAkt is decreased while NF-kB is activated. Please explain or discuss this phenomenon in the discussion.
4. Please indicate the specific phosphorylation site of Phospho-IRE1α, Phospho-eIF2α, Phospho-AKT, Phospho-IkBα and Phospho-NF-κB in the text.
1. There are many spelling mistakes or uncompleted sentences in the text and figures including supplementary figures. E.g. the font in lines 258-261 is different from the context words. The second “group” in line 316 can be deleted. The sentence “After 327 LPS treatment, WT (p<0.001, 1.59-fold) and Pgrmc1 KO (p<0.001, 1.53-fold) mice showed an increased compared to the normal state (Supplemental Figure S2)” in lines 327-329 is uncompleted. The sentence “Next, we measured in SH-SY5Y cells” in line 423, “AKktprotein” in line 539, and “showed an increased protein levels” in line 562 are also uncompleted. “diseased spinal cords and after traumatic brain injury” in line 587 can be rewritten as “spinal cord-associated diseases or traumatic brain injuries”.
2. There needs space between numbers and units throughout the text.
3. Please check and double-check spelling, punctuation and grammar of the text and figures.
Author Response
"Please see the attachment."
